# Association between bicarbonate levels and mortality among acute respiratory distress syndrome patients: An analysis based on Medical Information Mart for Intensive Care database

Junli Han[1]*, Lianghe Wang[2], Lingling Jin[1], Mingzhu Liu[1]

1 Department of Critical Care Medicine, The Second Affiliated Hospital of Xi'an Jiaotong University, Xi'an, P.R. China, 2 Department of Critical Care Medicine, Ngari Prefecture People's Hospital, Ngari Prefecture, P.R. China

* junlihan1102@163.com

## Abstract

### Objective

This study explored the association between serum bicarbonate levels and mortality risk among patients with acute respiratory distress syndrome (ARDS) admitted to the intensive care unit (ICU).

### Methods

This was a retrospective cohort study utilizing data extracted from the Medical Information Mart for Intensive Care (MIMIC)-IV database. Cox proportional hazards models and restricted cubic splines (RCS) were deployed for elucidating the association between the baseline bicarbonate levels and the risk of 28-day mortality while utilizing the Kaplan-Meier method to estimate survival curves, with hazard ratio (HR) and 95% confidence interval (CI). Subgroup analyses were conducted based on age, gender, Charlson Comorbidity Index (CCI) score, ARDS severity and bicarbonate administration.

### Results

Totally, 6,377 patients (15.38% deaths) were included. Baseline bicarbonate was significantly associated with 28-day mortality (HR: 0.98, 95% CI: 0.97–1.00, $P = 0.011$) in patients with ARDS. This association was particularly evident in female patients (HR: 1.16, 95% CI: 1.14–1.87, $P = 0.003$), those with a CCI of 2 or higher (HR: 1.27, 95% CI: 1.05–1.53, $P = 0.013$), among those with a $PaO_2/FiO_2$ ratio ranging from 200 to 300 mmHg (HR: 1.39, 95% CI: 1.08–1.78, $P = 0.011$), and those without bicarbonate administration (HR = 1.26, 95%CI: 1.07–1.48, $P = 0.004$), where bicarbonate levels falling below 23 mEq/L were linked to a heightened risk of not surviving the first 28

**Data availability statement:** The datasets generated and/or analyzed during the current study are available in the MIMIV-IV, https://physionet.org/content/mimiciv/3.0.

**Funding:** This study was supported by General Project of Shaanxi Provincial Department of Science and Technology- Social Development Field (grant number 2021SF-256). The funders had no role in study design, data collection and analysis, decision to publish, or preparation of the manuscript.

**Competing interests:** The authors have declared that no competing interests exist.

days in ARDS patients. RCS analysis revealed that the bicarbonate levels were non-linear associated with the 28-day mortality in ARDS patients (*P* for non-linear <0.001).

## Conclusion

Lower serum bicarbonate levels are significantly associated with an increased 28-day mortality risk in ARDS patients, with particular emphasis on female patients, those with higher CCI scores, and those with milder ARDS. Baseline bicarbonate levels of ARDS patients in ICU have certain clinical reference value for the development of clinical management and the assessment of prognostic risk during the ICU admission.

## Introduction

Acute respiratory distress syndrome (ARDS) is an acute inflammatory lung injury characterized by severe hypoxemic respiratory failure, bilateral opacities on chest imaging, and reduced lung compliance [1]. Each year, approximately 3 million people worldwide are affected by ARDS, accounting for about 10% of intensive care unit (ICU) admissions [2,3]. Mortality from ARDS remains a significant global health challenge [4]. The in-hospital mortality rate for patients with severe ARDS ranges from 46% to 60% [5]. Therefore, identifying factors associated with the risk of mortality in ARDS is crucial for effectively managing ARDS patients.

Bicarbonate is a crucial component of the human buffering system, playing a pivotal physiological role in maintaining acid-base homeostasis, supporting energy metabolism, regulating renal function, and facilitating digestive and secretory processes [6]. Previous clinical studies have revealed that ARDS is characterized by inflammatory pulmonary edema and impaired gas exchange, with endothelial injury serving as one of the pivotal mechanisms underlying its pathogenesis and progression. Endothelial injury drives the pathological process of ARDS by increasing microvascular permeability, disrupting the coagulation-fibrinolysis balance, and amplified inflammatory responses [7]. Evidence suggests that lower serum bicarbonate levels may be associated with endothelial dysfunction, potentially mediated through multiple pathophysiological pathways, including acidosis-induced endothelial injury, activation of inflammatory cascades, and microcirculatory impairment [8,9]. The bicarbonate buffer system serves as one of the most primary buffer systems for maintaining acid-base balance in the human body. When acidic substances increase, bicarbonate can combine with hydrogen ions to form carbon dioxide and water, thereby neutralizing the acids [10]. Excessive loss bicarbonate compromises the body's ability to neutralize acid, leading to accumulation of acidic substances in extracellular fluid and subsequent metabolic acidosis. Metabolic acidosis aggravates ARDS by disrupting the alveolar-capillary barrier, increasing endothelial injury and amplifying inflammatory responses [11,12]. The association between bicarbonate levels and the prognosis in clinically ill patients has been discussed. Du et al. [13] reported that baseline

bicarbonate levels were negatively associated with short-term mortality in patients with non-traumatic subarachnoid hemorrhage. A study derived from the MIMIC database similarly identified that low serum bicarbonate levels measured at ICU admission may serve as a biomarker for predicting both short- and long-term mortality in patients with acute aortic dissection (AAD) [14]. Moreover, baseline low bicarbonate levels have also been observed to be associated with poorer short-term outcomes in acute ischemic stroke patients admitted to the ICU [15]. However, the studies on the association between baseline bicarbonate levels and mortality risk in ARDS patients are lacking. This gap limits the application of bicarbonate in predicting the prognosis of ARDS patients in critical care.

Therefore, this study investigated the association between bicarbonate levels and 28-day mortality among ICU-admitted ARDS patients. Understanding this relationship may provide valuable insights into risk stratification and the potential role of bicarbonate levels as a prognostic marker, contributing to more clinical decision-making and tailored management strategies for ARDS patients.

## Methods

### Data source and patients

This study utilized a retrospective cohort design, including patients admitted between 2008 and 2019. Data were extracted from the Medical Information Mart for Intensive Care (MIMIC)-IV database (https://mimic.mit.edu/docs/iv/), a relational database containing real hospital stays for patients admitted to a tertiary academic medical center in Boston, MA, USA. MIMIC-IV provides comprehensive patient information during hospitalization, including laboratory measurements, medications administered, and documented vital signs, supporting diverse healthcare research with numerous improvements over its predecessor, MIMIC-III. Patients diagnosed with ARDS upon their first ICU admission were included. Exclusion criteria were as follows: age < 18 years, ICU stay <24 hours, missing serum creatinine information, end-stage renal disease, estimated glomerular filtration rate (eGFR) <15 ml/min/1.73m² at baseline (first 24 hours in ICU), hepatic failure, no bicarbonate measurement within 24 hours after ICU admission, no mechanical ventilation within 24 hours after ICU admission, or missing survival information. To access this database, one of the authors (Junli Han) obtained the necessary certification and subsequently extracted the relevant variables for our study (certification number: 13804349). The use of the MIMIC-IV database was approved by the review committee of Massachusetts Institute of Technology and Beth Israel Deaconess Medical Center; therefore, the ethics review committee of the Second Affiliated Hospital of Xi'an Jiaotong University waived the ethics approval statement and the requirement for informed consent for this study.

### Data collection

A comprehensive range of variables was collected and categorized into four major domains: demographic characteristics, clinical measurements, comorbidities, and treatments/interventions. Demographic characteristics included age, gender, race (Black, White, Others, or Unknown) and marital status (married, not married, or unknown), and insurance type (Medicaid, Medicare, or Others). Clinical measurements encompassed vital signs, including heart rate (beats per minute, bpm), systolic and diastolic blood pressure (mmHg), respiratory rate (breaths per minute, bpm), temperature (°C), oxygen saturation ($SpO_2$, %), and positive end-expiratory pressure (PEEP, mmHg). Laboratory values included white blood cell count (WBC, K/μL), platelet count (K/μL), hemoglobin (g/dL), creatinine (mg/dL), international normalized ratio (INR), prothrombin time (PT, seconds), blood urea nitrogen (BUN, mg/dL), lactate (mmol/L), bicarbonate (mEq/L), glucose (mg/dL), magnesium (mg/dL), calcium (mg/dL), sodium (mEq/L), chloride (mEq/L), and phosphate (mg/dL). Illness severity scores included the Sequential Organ Failure Assessment (SOFA), Simplified Acute Physiology Score II (SAPS II), Glasgow Coma Scale (GCS), and the Charlson Comorbidity Index (CCI). Additional parameters recorded were the arterial oxygen partial pressure to fractional inspired oxygen ratio ($PaO_2/FiO_2$, mmHg), urine output (mL/24h), weight (kg), and arterial blood gas values, including partial pressure of carbon dioxide ($PaCO_2$, mmHg), partial pressure of oxygen ($PaO_2$, mmHg), and pH. eGFR was categorized as <60, 60–90, or ≥90 mL/min/1.73 m². Comorbidities included diabetes, cancer, chronic kidney disease

(CKD), and heart failure, identified using the International Classification of Diseases (ICD-9/ICD-10) codes. Treatments and interventions included data on renal replacement therapy (RRT), vasopressor use, sodium bicarbonate administration, and sedatives (midazolam, propofol, and dexmedetomidine), as well as opioid usage (categorized by drug type: fentanyl, hydromorphone, meperidine, morphine, or multiple drugs) and antibiotic administration. All variables were systematically extracted from the MIMIC-IV database using predefined codes and item IDs to ensure consistency and accuracy.

## Measurement of ARDS and bicarbonate

ARDS was defined according to the Berlin criteria, which include the following: acute onset, a $PaO_2/FiO_2 \leq 300$ mmHg, PEEP ≥5 cm $H_2O$, bilateral infiltrates on chest radiograph, and the absence of heart failure. ARDS severity was classified based on the $PaO_2/FiO_2$ ratio: mild (200 mmHg < $PaO_2/FiO_2 \leq 300$ mmHg), moderate (100 mmHg < $PaO_2/FiO_2 \leq 200$ mmHg), and severe ($PaO_2/FiO_2 \leq 100$ mmHg) [16–18]. Baseline bicarbonate levels were categorized into two groups using the median value of 23 mEq/L as the cutoff.

## Outcome and follow-up

The outcome of this study was 28-day mortality. In-hospital follow-up data were recorded by hospital departments, while post-discharge information was tracked through records from the Social Security Administration. The follow-up period ended at the occurrence of 28-day mortality or 28 days after ICU discharge, whichever occurred first.

## Statistical analysis

Quantitative data were tested for normality using skewness and kurtosis, while homogeneity of variance was assessed using the Levene test. Normally distributed data were presented as a mean and standard deviation [Mean (±SD)] and compared between groups using the t-test for equal variances or the adjusted t' test for unequal variances. Non-normally distributed data were described as median and interquartile range [M ($Q_1$, $Q_3$)] and compared using the Wilcoxon rank-sum test. Categorical data were presented as counts and proportions [n (%)] and compared using the chi-square test or Fisher's exact test. Variables with missing values exceeding 20% were excluded, while those with missing values ≤20% were imputed using a multiple imputation method. Missing values were handled using the mice () function from R's mice package (v 4.3.1), which starts with a data frame containing missing values and generates multiple (default of five) complete datasets by imputing the missing values. Each completed dataset is then analyzed separately. Considering the potential influence of this imputation method on the relationship between exposure and outcomes, sensitivity analysis was performed by comparing results before and after imputations (S1 Table). The distribution differences of all variables before and after imputation were not statistically significant (all $P > 0.05$), thus, the imputation of missing data conducted in this study supported the robustness of the study results.

Univariate Cox proportional hazards model was used to preliminarily screen the covariates with statistical association with the 28-day mortality risk among all variables, and the non-statistically significant variables were excluded to reduce the dimensions of subsequent multivariate Cox proportional hazards model analysis. In Model 1, only univariate analyses were performed for each variable. Model 2 included all variables identified from the previous step, which were retained through a bidirectional stepwise regression process. Univariate and multivariable Cox proportional hazards models were used to explore the association between baseline bicarbonate levels and the risk of 28-day mortality in ARDS patients. The univariate model analyzed associations without adjusting for any factors, while the multivariate model adjusted for covariates screened out in univariate model to clarify the association between baseline bicarbonate levels and the risk of 28-day mortality in patients with ARDS. Subgroup analyses were conducted to evaluate the association between baseline bicarbonate levels and 28-day mortality across different patient groups. The subgroups were defined based on age (<65 years and ≥65 years), gender (male and female), CCI, ARDS severity, and with or without bicarbonate administration. The Kaplan-Meier survival curve illustrates the relationship between bicarbonate levels and the 28-day survival probability in

ARDS patients. Moreover, the restricted cubic splines (RCS) curve was performed to assess the nonlinear association between baseline bicarbonate levels and the risk of 28-day mortality. The evaluation metrics included the hazard ratio (HR) and its corresponding 95% confidence interval (CI). The confidence level was set at alpha = 0.05. Data cleaning was performed using Python 3.9.12, while missing value analysis, data imputation, sensitivity analysis, comparative testing, statistical modeling, and visualization were conducted using R version 4.3.1 (2023-06-16 ucrt).

## Results

### Study selection process and characteristics of included patients

The flowchart (Fig 1) illustrates the patient selection process for the study from the MIMIC-IV database. Initially, 8,316 patient records diagnosed with ARDS upon their first ICU admission were identified. A total of 1,939 records were excluded based on the inclusion and exclusion criteria. After applying these exclusion criteria, 6,377 patients remained in the final study sample. Among these, 5,396 patients survived, while 981 patients died within 28 days of ICU admission. The mean age of the included patients was 62.15 (±15.01) years. Among them, 2,271 (35.61%) were female, and 4,106 (64.39%) were male. The baseline bicarbonate level in the study was 22.86 (±4.35) mEq/L. Significant differences were observed between survivors and non-survivors in the following variables: age, gender, race, insurance type, marital status, malignant cancer, CKD, $PaO_2/FiO_2$, urine output (24h), weight, heart rate, diastolic blood pressure, respiratory rate, $SpO_2$, PEEP, $FiO_2$, SOFA, SAPSII, GCS, CCI, platelet, creatinine, INR, PT, BUN, lactate, $PaCO_2$, $PaO_2$, pH, magnesium, calcium, sodium, chloride, phosphate, glucose, RRT, vasopressor use, sodium bicarbonate use, midazolam, propofol, dexmedetomidine, opioid use, antibiotic use, eGFR, ICU length of stay, and bicarbonate levels. Table 1 summarizes the characteristics of the included patients.

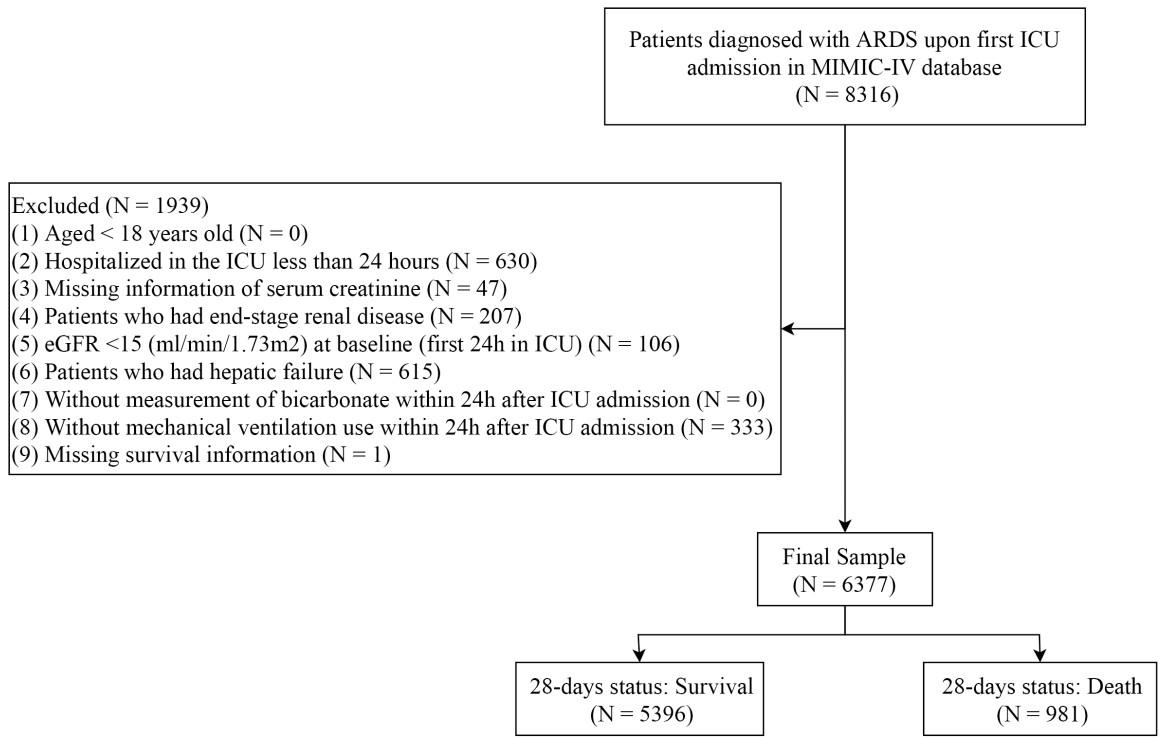

**Fig 1. The patient selection process for the study.**

### Association between baseline bicarbonate levels and 28-day mortality risk in ARDS patients

Table 2 presents the association between baseline bicarbonate levels and the risk of 28-day mortality. Baseline bicarbonate was significantly associated with 28-day mortality (HR: 0.98, 95% CI: 0.97–1.00, $P=0.011$). When bicarbonate levels were categorized, patients with bicarbonate <23 mEq/L was associated with an increased risk of 28-day mortality compared to those with bicarbonate ≥23 mEq/L (HR: 1.22, 95% CI: 1.05–1.41, $P=0009$). The Kaplan-Meier curve (Fig 2) shows a higher survival probability is typically observed in patients with higher bicarbonate levels, while those with lower bicarbonate levels may show a significantly lower survival probability. Moreover, the RCS curve analysis revealed a significant nonlinear relationship between the bicarbonate levels and 28-day mortality in patients with ARDS ($P$ for non-linear <0.001) (Fig 3).

### Subgroup analysis of the association between baseline bicarbonate levels and 28-day mortality risk in ARDS patients

In female patients (HR: 1.16, 95% CI: 1.14–1.87, $P=0.003$), in those with CCI ≥ 2 (HR: 1.27, 95% CI: 1.05–1.53, $P=0.013$), in patients with $PaO_2/FiO_2$ between 200−300 mmHg (HR: 1.39, 5% CI: 1.08–1.78, $P=0.011$), in patients without bicarbonate administration (HR: 1.26, 95%CI: 1.07–1.48, $P=0.004$), and patients without bicarbonate administration (HR = 1.26, 95%CI: 1.07–1.48, $P=0.004$), bicarbonate levels <23 mEq/L were associated with an increased risk of 28-day mortality in ARDS patients. Table 3 illustrates the subgroup analysis, highlighting the association between baseline bicarbonate levels and the risk of 28-day mortality in different patient subgroups.

## Discussion

To the best of our knowledge, this was the first study conducted based on MIMIV-IV database to explore the association between baseline bicarbonate levels and 28-day mortality in patients with ARDS. The analysis revealed that lower serum bicarbonate levels were significantly associated with an increased risk of 28-day mortality in patients with ARDS, particularly in female patients, those with higher CCI scores, those with milder ARDS, and those without bicarbonate administration.

Several epidemiological studies have explored the association between bicarbonate levels and the mortality risk among non-ARDS patients. First, in a study of 3,281 patients who underwent primary colorectal cancer resection, multivariable analysis of data from 2,223 patients revealed a significant association between serum bicarbonate levels and perioperative mortality, but no significant association with five-year survival rates [19]. Similarly, large-scale studies in renal disease populations have yielded important insights. The multinational Dialysis Outcomes and Practice Patterns Study (DOPPS), including over 7,000 hemodialysis patients, revealed that bicarbonate levels higher than 27 mEq/L and lower than 17 mEq/L were associated with an increased risk of mortality [20]. This pattern was further supported by a South Korean study of 1,159 dialysis patients (median follow-up 37 months), where bicarbonate levels ≥24 mEq/L predicted higher mortality [21]. Notably, analyses of renal populations consistently reveal this dual risk pattern. A study of 41,749 patients with advanced kidney disease found that both hypo- and hyperbicarbonatemia were independently associated with elevated all-cause mortality [22]. A study analyzing data from 121,351 end-stage renal disease patients (including 10,400 peritoneal dialysis and 110,951 hemodialysis patients) between 2001 and 2006 found that serum bicarbonate levels <22 mEq/L were significantly associated with both all-cause mortality and cardiovascular mortality [23]. However, the National Health and Nutrition Examination Survey study found that low serum bicarbonate levels did not demonstrate strong predictive power for mortality in non-CKD patients [24]. Beyond renal disease, this association persists in other critical conditions. The findings from the retrospective, propensity-matched cohort study using the Tri-NetX database suggest a strong association between low serum bicarbonate levels and increased mortality risk [25]. In a study of 165 patients with ischemic cardiogenic shock (118 males, mean age 68.4 years), baseline serum bicarbonate

**Table 1. Characteristics of the included patients.**

| Variables | Total (N = 6377) | 28-day survival | | Statistics | P |
|---|---|---|---|---|---|
| | | Yes (N = 5396) | No (N = 981) | | |
| **Demographic information** | | | | | |
| Age, years, Mean (±SD) | 62.15 (±15.01) | 61.31 (±14.86) | 66.79 (±15.02) | t = −10.623 | <0.001 |
| Gender, n (%) | | | | $\chi^2$ = 45.580 | <0.001 |
| Female | 2271 (35.61) | 1828 (33.88) | 443 (45.16) | | |
| Male | 4106 (64.39) | 3568 (66.12) | 538 (54.84) | | |
| Race, n (%) | | | | $\chi^2$ = 64.334 | <0.001 |
| Black | 436 (6.84) | 357 (6.62) | 79 (8.05) | | |
| Others | 684 (10.73) | 587 (10.88) | 97 (9.89) | | |
| Unknown | 1009 (15.82) | 774 (14.34) | 235 (23.96) | | |
| White | 4248 (66.61) | 3678 (68.16) | 570 (58.1) | | |
| Marital status, n (%) | | | | $\chi^2$ = 108.984 | <0.001 |
| Married | 3095 (48.53) | 2724 (50.48) | 371 (37.82) | | |
| No married | 2565 (40.22) | 2152 (39.88) | 413 (42.1) | | |
| Unknown | 717 (11.24) | 520 (9.64) | 197 (20.08) | | |
| Insurance type, n (%) | | | | $\chi^2$ = 47.937 | <0.001 |
| Medicaid | 500 (7.84) | 426 (7.89) | 74 (7.54) | | |
| Medicare | 2504 (39.27) | 2023 (37.49) | 481 (49.03) | | |
| Others | 3373 (52.89) | 2947 (54.61) | 426 (43.43) | | |
| **Comorbidities** | | | | | |
| Diabetes, n (%) | | | | $\chi^2$ = 3.188 | 0.074 |
| No | 4612 (72.32) | 3879 (71.89) | 733 (74.72) | | |
| Yes | 1765 (27.68) | 1517 (28.11) | 248 (25.28) | | |
| Malignant cancer, n (%) | | | | $\chi^2$ = 109.651 | <0.001 |
| No | 5613 (88.02) | 4848 (89.84) | 765 (77.98) | | |
| Yes | 764 (11.98) | 548 (10.16) | 216 (22.02) | | |
| CKD, n (%) | | | | $\chi^2$ = 15.972 | <0.001 |
| No | 5695 (89.31) | 4855 (89.97) | 840 (85.63) | | |
| Yes | 682 (10.69) | 541 (10.03) | 141 (14.37) | | |
| **Vital signs** | | | | | |
| Pao2/Fio2, n (%) | | | | $\chi^2$ = 148.408 | <0.001 |
| ≤100 | 932 (14.62) | 682 (12.64) | 250 (25.48) | | |
| 100-200 | 2390 (37.48) | 1987 (36.82) | 403 (41.08) | | |
| 200-300 | 3055 (47.91) | 2727 (50.54) | 328 (33.44) | | |
| Urine output 24h, mL, M (Q$_1$, Q$_3$) | 1705 (1132-2420) | 1772.5 (1233-2465) | 1140 (642-1973) | W = 3519340 | <0.001 |
| Weight, kg, Mean (±SD) | 86.07 (±20.94) | 87.18 (±20.76) | 79.95 (±20.86) | t = 10.020 | <0.001 |
| Heart rate, bpm, Mean (±SD) | 88.87 (±19.58) | 87.79 (±18.89) | 94.84 (±22.07) | t' = −9.402 | <0.001 |
| Systolic, mmHg, Mean (±SD) | 120.80 (±23.36) | 120.65 (±22.78) | 121.66 (±26.33) | t' = −1.131 | 0.258 |
| Diastolic, mmHg, Mean (±SD) | 65.81 (±15.87) | 65.51 (±15.47) | 67.51 (±17.85) | t' = −3.300 | 0.001 |
| Respiratory rate, bpm, Mean (±SD) | 18.44 (±5.64) | 17.99 (±5.49) | 20.93 (±5.80) | t' = −14.725 | <0.001 |
| Temperature, °C, Mean (±SD) | 36.63 (±0.75) | 36.64 (±0.74) | 36.62 (±0.84) | t' = 0.460 | 0.646 |
| SpO2, %, Mean (±SD) | 97.80 (±3.01) | 97.99 (±2.88) | 96.77 (±3.46) | t' = 10.435 | <0.001 |
| PEEP, mmHg, Mean (±SD) | 6.17 (±2.45) | 6.06 (±2.32) | 6.80 (±3.02) | t' = −7.370 | <0.001 |
| FiO2, %, Mean (±SD) | 82.16 (±23.80) | 83.19 (±23.52) | 76.50 (±24.50) | t' = 7.915 | <0.001 |

*(Continued)*

**Table 1.** (Continued)

| Variables | Total (N = 6377) | 28-day survival | | Statistics | P |
|---|---|---|---|---|---|
| | | Yes (N = 5396) | No (N = 981) | | |
| **Illness severity scores** | | | | | |
| SOFA, score, Mean (±SD) | 6.86 (±3.61) | 6.35 (±3.27) | 9.69 (±4.04) | t' = −24.504 | <0.001 |
| SAPSII, score, Mean (±SD) | 39.88 (±14.26) | 37.68 (±12.80) | 51.99 (±15.75) | t' = −26.895 | <0.001 |
| GCS, score, Mean (±SD) | 11.44 (±4.21) | 11.87 (±3.92) | 9.10 (±4.95) | t' = 16.566 | <0.001 |
| CCI, score, Mean (±SD) | 2.09 (±2.12) | 1.90 (±1.96) | 3.15 (±2.60) | t' = −14.362 | <0.001 |
| **Laboratory value** | | | | | |
| WBC, K/uL, M (Q$_1$, Q$_3$) | 12.4 (8.8-16.6) | 12.3 (8.9-16.3) | 12.7 (8.3-18.3) | W = 2555430.5 | 0.085 |
| Platelet, K/uL, Mean (±SD) | 201.17 (±109.08) | 198.63 (±105.11) | 215.19 (±127.90) | t' = −3.829 | <0.001 |
| Hemoglobin, g/dL, Mean (±SD) | 10.77 (±2.19) | 10.78 (±2.15) | 10.71 (±2.43) | t' = 0.904 | 0.366 |
| Creatinine, mg/dL, Mean (±SD) | 1.05 (±0.61) | 1.01 (±0.56) | 1.27 (±0.80) | t' = −9.585 | <0.001 |
| eGFR, n (%) | | | | χ² = 232.241 | <0.001 |
| <60 | 1358 (21.3) | 976 (18.09) | 382 (38.94) | | |
| 60-90 | 2129 (33.39) | 1823 (33.78) | 306 (31.19) | | |
| ≥90 | 2890 (45.32) | 2597 (48.13) | 293 (29.87) | | |
| INR, ratio, M (Q$_1$, Q$_3$) | 1.3 (1.2-1.5) | 1.3 (1.2-1.5) | 1.3 (1.16-1.6) | W = 2436837.5 | <0.001 |
| PT, seconds, M (Q$_1$, Q$_3$) | 14.5 (13-16.3) | 14.4 (13-16.2) | 14.7 (12.9-18) | W = 2431421 | <0.001 |
| BUN, mg/dL, M (Q$_1$, Q$_3$) | 17 (12-24) | 16 (12-22) | 23 (16-35) | W = 1724344.5 | <0.001 |
| Lactate, mmol/L, M (Q$_1$, Q$_3$) | 1.9 (1.4-2.7) | 1.9 (1.4-2.6) | 2.1 (1.4-3.5) | W = 2291905 | <0.001 |
| PaCO2, mmHg, M (Q$_1$, Q$_3$) | 42 (37-48) | 42 (38-48) | 40 (34-49) | W = 2933198.5 | <0.001 |
| PaO2, mmHg, Mean (±SD) | 153.19 (±75.25) | 159.73 (±76.16) | 117.23 (±58.09) | t' = 20.004 | <0.001 |
| PH, Mean (±SD) | 7.35 (±0.10) | 7.36 (±0.09) | 7.32 (±0.13) | t' = 8.798 | <0.001 |
| Magnesium, mg/dL, Mean (±SD) | 2.04 (±0.50) | 2.06 (±0.51) | 1.94 (±0.46) | t' = 7.059 | <0.001 |
| Calcium, mg/dL, M (Q$_1$, Q$_3$) | 8.1 (7.7-8.5) | 8.12 (7.76-8.5) | 8.1 (7.5-8.7) | W = 2775954 | 0.015 |
| Sodium, mEq/L, Mean (±SD) | 137.48 (±4.89) | 137.26 (±4.55) | 138.71 (±6.29) | t' = −6.916 | <0.001 |
| Chloride, mEq/L, M (Q$_1$, Q$_3$) | 106 (102-109) | 106 (103-109) | 105 (101-109) | W = 2788177.5 | 0.008 |
| Phosphate, mg/dL, M (Q$_1$, Q$_3$) | 3.5 (3-4.2) | 3.5 (3-4.2) | 3.7 (2.8-4.7) | W = 2470167 | 0.001 |
| Glucose, mg/dL, M (Q$_1$, Q$_3$) | 141 (116-175) | 140 (116-172) | 145 (115-195) | W = 2482402.5 | 0.002 |
| **Treatments** | | | | | |
| RRT, n (%) | | | | χ² = 28.508 | <0.001 |
| No | 6281 (98.49) | 5334 (98.85) | 947 (96.53) | | |
| Yes | 96 (1.51) | 62 (1.15) | 34 (3.47) | | |
| Vasopressor, n (%) | | | | χ² = 23.387 | <0.001 |
| No | 2600 (40.77) | 2269 (42.05) | 331 (33.74) | | |
| Yes | 3777 (59.23) | 3127 (57.95) | 650 (66.26) | | |
| Sodium bicarbonate use, n (%) | | | | χ² = 138.550 | <0.001 |
| No | 5917 (92.79) | 5095 (94.42) | 822 (83.79) | | |
| Yes | 460 (7.21) | 301 (5.58) | 159 (16.21) | | |
| Midazolam, n (%) | | | | χ² = 183.513 | <0.001 |
| No | 4707 (73.81) | 4155 (77) | 552 (56.27) | | |
| Yes | 1670 (26.19) | 1241 (23) | 429 (43.73) | | |
| Propofol, n (%) | | | | χ² = 350.704 | <0.001 |
| No | 1124 (17.63) | 745 (13.81) | 379 (38.63) | | |
| Yes | 5253 (82.37) | 4651 (86.19) | 602 (61.37) | | |

*(Continued)*

**Table 1.** (Continued)

| Variables | Total (N = 6377) | 28-day survival | | Statistics | P |
|---|---|---|---|---|---|
| | | Yes (N = 5396) | No (N = 981) | | |
| Dexmedetomidine, n (%) | | | | $\chi^2 = 91.434$ | <0.001 |
| No | 5580 (87.5) | 4630 (85.8) | 950 (96.84) | | |
| Yes | 797 (12.5) | 766 (14.2) | 31 (3.16) | | |
| Opioid, n (%) | | | | – | <0.001 |
| No use | 834 (13.08) | 643 (11.92) | 191 (19.47) | | |
| Fentanyl | 3491 (54.74) | 2866 (53.11) | 625 (63.71) | | |
| Hydromorphone | 257 (4.03) | 232 (4.3) | 25 (2.55) | | |
| Meperidine | 8 (0.13) | 8 (0.15) | 0 (0) | | |
| Morphine | 832 (13.05) | 776 (14.38) | 56 (5.71) | | |
| More than two types of drugs | 955 (14.98) | 871 (16.14) | 84 (8.56) | | |
| Antibiotics, n (%) | | | | $\chi^2 = 20.740$ | <0.001 |
| No | 1145 (17.96) | 918 (17.01) | 227 (23.14) | | |
| Yes | 5232 (82.04) | 4478 (82.99) | 754 (76.86) | | |
| **Exposure and outcome** | | | | | |
| Length of ICU stay, days, M (Q₁, Q₃) | 3.15 (1.69-6.97) | 3.04 (1.55-6.59) | 4.36 (2.04-8.58) | W = 2213477 | <0.001 |
| Follow time, days, Mean (±SD) | 24.98 (±7.57) | 28.00 (±0.00) | 8.34 (±6.72) | t' = 91.663 | <0.001 |
| Bicarbonate, mEq/L, Mean (±SD) | 22.86 (±4.35) | 23.06 (±4.08) | 21.71 (±5.47) | t' = 7.371 | <0.001 |
| Baseline bicarbonate level, n (%) | | | | $\chi^2 = 68.370$ | <0.001 |
| ≥23 | 3523 (55.25) | 3100 (57.45) | 423 (43.12) | | |
| <23 | 2854 (44.75) | 2296 (42.55) | 558 (56.88) | | |

Notes: CKD: chronic kidney disease; PaO$_2$: arterial oxygen partial pressure; FiO$_2$: arterial oxygen partial pressure; SpO$_2$: oxygen saturation; PEEP: positive end-expiratory pressure; SOFA: Sequential Organ Failure Assessment; SAPS II: Simplified Acute Physiology Score II; GCS: Glasgow Coma Scale; CCI: Charlson Comorbidity Index; WBC: white blood cell; INR: international normalized ratio; PT: prothrombin time; BUN: blood urea nitrogen; PaCO$_2$: partial pressure of carbon dioxide; RRT: renal replacement therapy; eGFR: estimated glomerular filtration rate; ICU: intensive care unit; SD: Standard Deviation; M: Median; Q₁: 1st Quartile; Q₃: 3st Quartile; t: Student's t test; t': Satterthwaite t test; W: Wilcoxon rank sum test; $\chi^2$: Chi-square test; -: Fisher's exact test.

**Table 2.** Association between baseline bicarbonate levels and 28-day mortality risk in ARDS patients.

| Variables | Model 1 | | Model 2 | |
|---|---|---|---|---|
| | HR (95% CI) | P | HR (95% CI) | P |
| Baseline bicarbonate levels | 0.93 (0.92–0.94) | <0.001 | 0.98 (0.97–1.00) | 0.011 |
| Baseline bicarbonate levels | | | | |
| ≥23 | Ref | | Ref | |
| <23 | 1.71 (1.51–1.95) | <0.001 | 1.22 (1.05–1.41) | 0.009 |

Notes: ARDS: Acute respiratory distress syndrome; HR, hazard ratio; CI: Confidence intervals; Ref: reference;

Model 1: Crude model;

Model 2 adjusted age, gender, race, marital status, weight, diastolic blood pressure, respiratory rate, SOFA, SAPSII, CCI, platelet, lactate, magnesium, sodium, chloride, RRT, sodium bicarbonate use, propofol, dexmedetomidine, opioid, and antibiotics.

levels were found to independently predict 28-day and 365-day mortality [26]. Likewise, among 4,048 acute ischemic stroke patients in ICU settings, bicarbonate levels <21 mEq/L and declining trends were significantly associated with 30-day mortality [15]. Furthermore, in ICU patients with acute aortic dissection, levels <22 mmol/L predicted increased mortality at 30 days, 90 days, 1 year, and 5 years [14]. Importantly, our study represents the first suggest that baseline

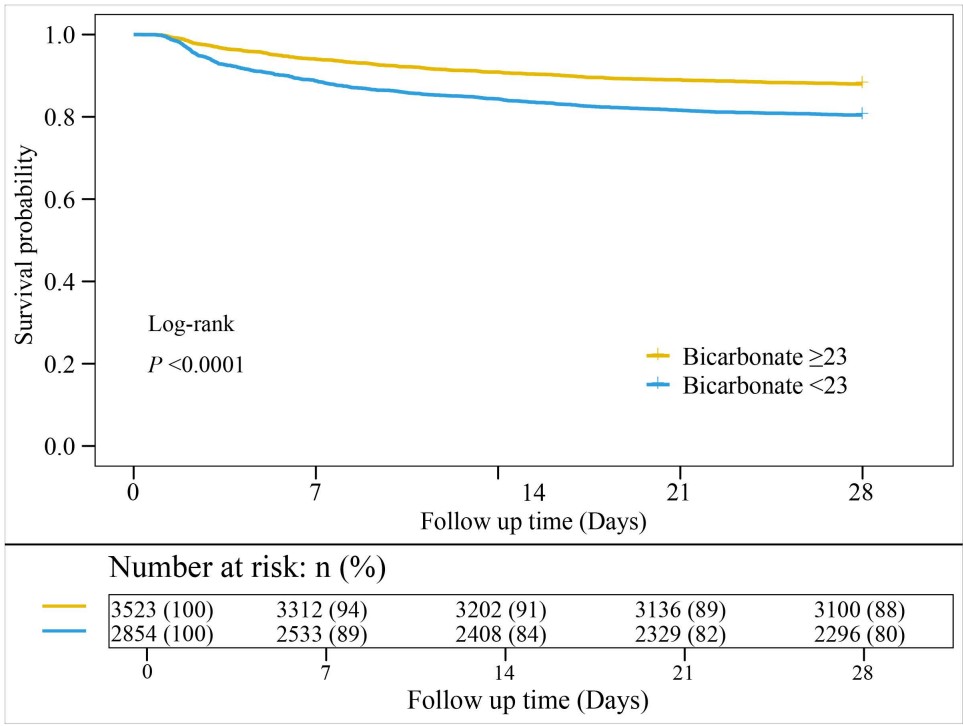

**Fig 2. The relationship between bicarbonate levels and the 28-day survival probability in ARDS patients.**

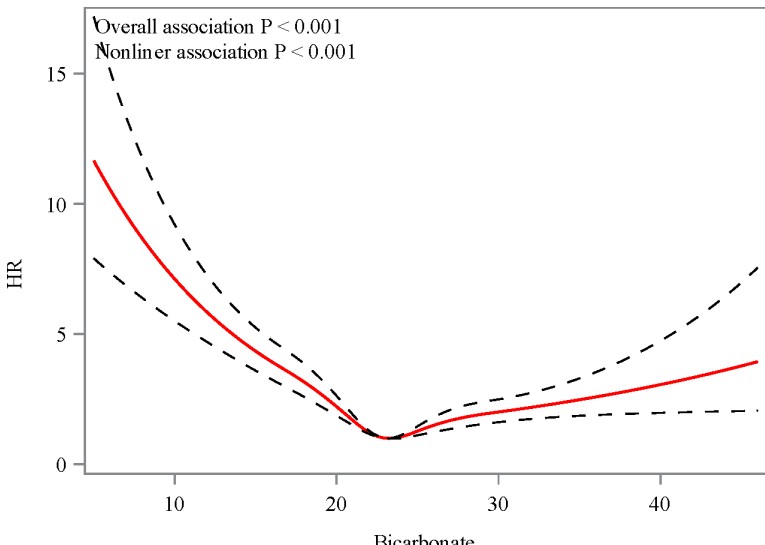

**Fig 3. The restricted cubic spline analysis of bicarbonate levels and the 28-day mortality in ARDS patients.**

low serum bicarbonate levels are associated with elevated 28-day mortality risk specifically in ARDS patients. These collective findings highlight the importance of monitoring bicarbonate levels in ARDS patients, as maintaining optimal levels could potentially improve outcomes by reducing mortality risk.

**Table 3.** Subgroup analysis of the association between baseline bicarbonate levels and 28-day mortality risk in ARDS patients.

| Subgroups (Outcome/Total) | HR (95% CI) | P |
|---|---|---|
| Age < 65 years (N = 406/3384) | | |
| Baseline bicarbonate levels | 0.99 (0.97–1.01) | 0.457 |
| Baseline bicarbonate levels | | |
| ≥23 | Ref | |
| <23 | 1.27 (1.01–1.60) | 0.039 |
| Age ≥ 65 years (N = 575/2993) | | |
| Baseline bicarbonate levels | 0.96 (0.94–0.99) | 0.001 |
| Baseline bicarbonate levels | | |
| ≥23 | Ref | |
| <23 | 1.27 (1.04–1.56) | 0.020 |
| Female (N = 443/2271) | | |
| Baseline bicarbonate levels | 0.97 (0.95–0.99) | 0.016 |
| Baseline bicarbonate levels | | |
| ≥23 | Ref | |
| <23 | 1.46 (1.14–1.87) | 0.003 |
| Male (N = 538/4106) | | |
| Baseline bicarbonate levels | 0.98 (0.96–1.00) | 0.101 |
| Baseline bicarbonate levels | | |
| ≥23 | Ref | |
| <23 | 1.10 (0.91–1.35) | 0.322 |
| CCI < 2 (N = 324/3186) | | |
| Baseline bicarbonate levels | 0.99 (0.97–1.02) | 0.625 |
| Baseline bicarbonate levels | | |
| ≥23 | Ref | |
| <23 | 1.19 (0.92–1.54) | 0.177 |
| CCI ≥ 2 (N = 657/3191) | | |
| Baseline bicarbonate levels | 0.97 (0.95–0.99) | 0.001 |
| Baseline bicarbonate levels | | |
| ≥23 | Ref | |
| <23 | 1.27 (1.05–1.53) | 0.013 |
| Pao2/Fio2 ≤ 200 (N = 653/3322) | | |
| Baseline bicarbonate levels | 0.98 (0.97–1.00) | 0.096 |
| Baseline bicarbonate levels | | |
| ≥23 | Ref | |
| <23 | 1.12 (0.93–1.36) | 0.217 |
| Pao2/Fio2 = 200–300 (N = 328/3055) | | |
| Baseline bicarbonate levels | 0.97 (0.95–1.00) | 0.070 |
| Baseline bicarbonate levels | | |
| ≥23 | Ref | |
| <23 | 1.39 (1.08–1.78) | 0.011 |
| Non-bicarbonate administration (N = 822/5917) | | |
| Baseline bicarbonate level | 0.98 (0.96–0.99) | 0.006 |
| Baseline bicarbonate level | | |
| ≥23.00 | Ref | |
| <23.00 | 1.26 (1.07–1.48) | 0.004 |

*(Continued)*

**Table 3.** (Continued)

| Subgroups (Outcome/Total) | HR (95% CI) | *P* |
|---|---|---|
| Bicarbonate administration (N = 159/460) | | |
| Baseline bicarbonate level | 1.01 (0.97–1.05) | 0.687 |
| Baseline bicarbonate level | | |
| ≥23.00 | Ref | |
| <23.00 | 0.88 (0.56–1.38) | 0.575 |

Notes: ARDS: Acute respiratory distress syndrome; HR, hazard ratio; CI: Confidence intervals; Ref: reference; CCI: Charles Comorbidity Index; PaO$_2$: arterial oxygen partial pressure; FiO$_2$: arterial oxygen partial pressure;

Adjusted age, gender, race, marital status, weight, diastolic blood pressure, respiratory rate, SOFA, SAPSII, CCI, platelet, lactate, magnesium, sodium, chloride, RRT, sodium bicarbonate use, propofol, dexmedetomidine, opioid, and antibiotics.

The association between baseline low serum bicarbonate levels and an increased risk of 28-day mortality in ICU patients with ARDS could be attributed to several potential mechanisms. Low serum bicarbonate levels typically indicate metabolic acidosis [27], which can exacerbate inflammation, impair cellular function, and worsen organ dysfunction [28], ultimately increasing the risk of mortality in ARDS patients. Metabolic acidosis, reflected by low bicarbonate levels, may contribute to endothelial dysfunction [7], a key pathophysiological mechanism in ARDS [29]. This dysfunction can lead to increased vascular permeability, further pulmonary edema, and respiratory failure [30]. Low bicarbonate levels can directly affect cardiovascular function, leading to decreased cardiac output and increased risk of arrhythmias [25], contributing to poor outcomes in critically ill patients.

The analysis indicated that lower serum bicarbonate levels were significantly linked to a higher risk of 28-day mortality in patients with ARDS, especially among females, those with higher CCI scores, and individuals with less severe ARDS. Women may have a different physiological response to metabolic acidosis or altered acid-base regulation compared to men, which could make them more susceptible to the adverse effects of low bicarbonate levels [31,32], leading to higher mortality risk in ARDS. In addition, gender may influence the development of severe diseases [33]. A higher CCI indicates the presence of multiple comorbidities [34], which can further complicate the management of ARDS and increase the burden on the body. Comorbidities often associated with higher CCI scores, like diabetes or renal disease, can contribute to a heightened inflammatory state, which is further worsened by acidosis [22,25]. We hypothesize that the differing outcomes between patients with mild and severe ARDS may be attributed to variations in their compensatory mechanisms for acid-base balance [35]. These findings underscore the importance of monitoring bicarbonate levels in the management of ARDS patients, particularly in specific subgroups, as it may help identify those at higher risk of mortality and guide more personalized treatment strategies. Of course, the underlying pathophysiological mechanisms linking bicarbonate levels to the outcomes of ARDS with different characteristics require further investigation through dedicated mechanistic studies.

The finding that lower bicarbonate levels are associated with an increased risk of 28-day mortality in ARDS patients has important clinical implications. In ARDS patients, abnormal bicarbonate levels may signal underlying metabolic dysfunction that can exacerbate respiratory and circulatory instability, contributing to poorer outcomes [36,37]. Clinically, this means that monitoring bicarbonate levels in ARDS patients-especially those with comorbidities or milder forms of the disease-may help identify individuals at higher risk of mortality. For these high-risk patients, closer surveillance and timely interventions to correct acid-base imbalances could improve prognosis. For example, adjustments to respiratory support, such as optimizing ventilation strategies or administering bicarbonate supplements, might be necessary to restore metabolic balance and prevent further deterioration. By integrating bicarbonate monitoring into routine care, clinicians can better tailor interventions, potentially reducing mortality risk and improving patient outcomes in this critically ill population.

This study has several strengths and limitations. One of its major strengths was that it is the first to investigate the association between baseline bicarbonate levels and short-term mortality risk in critically ill ARDS patients, using a

large-scale cohort. Baseline bicarbonate level, as a potential prognostic biomarker for ARDS, holds dual clinical significance: on the one hand, lower bicarbonate levels reflect the metabolic acidosis and is associated with multi-organ dysfunction; on the other hand, lower bicarbonate levels can provide a reference for the stratification of mortality risk and help guide more effective clinical management of patients with ARDS in the ICU. However, we must acknowledge that our study has some objective limitations. First, as an observational, retrospective cohort study based on the MIMIC-IV database, our findings cannot establish a causal relationship between baseline bicarbonate levels and 28-day mortality risk in ARDS patients. Lower bicarbonate levels may serve as a prognostic marker rather than an independent risk factor for short-term outcomes. Second, being a single-center study, our results inevitably suffer from limited generalizability due to population heterogeneity, institution-specific practices, and insufficient data diversity. Therefore, caution was warranted when extrapolating these findings to patients from different regions, ethnicities, socioeconomic backgrounds, or healthcare settings. Third, although we adjusted for numerous potential confounders, residual bias may persist due to unmeasured variables and inherent limitations in MIMIC-IV data availability. Fourth, bicarbonate measurements were only available for ICU-admitted patients, restricting our ability to assess how post-ICU fluctuations in bicarbonate levels might influence mortality. This could affect the precision of our conclusions. Considering these limitations, further validation through well-designed multicenter prospective studies or randomized controlled trials (RCTs) is essential to confirm these associations and evaluate their broader clinical applicability.

## Conclusion

Based on the MIMIC-IV database, the analysis observed a significant association between lower serum bicarbonate levels and an elevated risk of 28-day mortality in ARDS patients. These findings suggest the potential benefit of closely monitoring and managing bicarbonate levels as part of the overall clinical strategy to reduce mortality risk. Further research is needed to explore whether correcting bicarbonate imbalances could directly contribute to improved patient outcomes in ARDS.

## Supporting information

**S1 Table. Sensitivity analysis results before and after imputations.**
(DOCX)

## Author contributions

**Conceptualization:** Junli Han.

**Data curation:** Lianghe Wang, Lingling Jin, Mingzhu Liu.

**Formal analysis:** Lianghe Wang, Lingling Jin, Mingzhu Liu.

**Funding acquisition:** Junli Han.

**Investigation:** Lianghe Wang, Lingling Jin, Mingzhu Liu.

**Methodology:** Lianghe Wang, Lingling Jin, Mingzhu Liu.

**Writing – original draft:** Junli Han.

**Writing – review & editing:** Junli Han.

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
