## [Decision Letter · Decision Letter 0]

6 Feb 2025

PONE-D-24-58051Association between bicarbonate levels and mortality among acute respiratory distress syndrome patients: An analysis based on Medical Information Mart for Intensive Care databasePLOS ONE

Dear Dr. Han,

Thank you for submitting your manuscript to PLOS ONE. After careful consideration, we feel that it has merit but does not fully meet PLOS ONE’s publication criteria as it currently stands. Therefore, we invite you to submit a revised version of the manuscript that addresses the points raised during the review process. We appreciate your contribution to this important area of research. While your study addresses a relevant clinical question, several key areas require further clarification and methodological refinement.

If applicable, we recommend that you deposit your laboratory protocols in protocols.io to enhance the reproducibility of your results. Protocols.io assigns your protocol its own identifier (DOI) so that it can be cited independently in the future. For instructions see: https://journals.plos.org/plosone/s/submission-guidelines#loc-laboratory-protocols . Additionally, PLOS ONE offers an option for publishing peer-reviewed Lab Protocol articles, which describe protocols hosted on protocols.io. Read more information on sharing protocols at https://plos.org/protocols?utm_medium=editorial-email&utm_source=authorletters&utm_campaign=protocols.

We look forward to receiving your revised manuscript.

Kind regards,

Reema Karasneh

Academic Editor

PLOS ONE

Journal requirements:   When submitting your revision, we need you to address these additional requirements. 1. Please ensure that your manuscript meets PLOS ONE's style requirements, including those for file naming. The PLOS ONE style templates can be found at https://journals.plos.org/plosone/s/file?id=wjVg/PLOSOne_formatting_sample_main_body.pdf and https://journals.plos.org/plosone/s/file?id=ba62/PLOSOne_formatting_sample_title_authors_affiliations.pdf. 2. PLOS requires an ORCID iD for the corresponding author in Editorial Manager on papers submitted after December 6th, 2016. Please ensure that you have an ORCID iD and that it is validated in Editorial Manager. To do this, go to ‘Update my Information’ (in the upper left-hand corner of the main menu), and click on the Fetch/Validate link next to the ORCID field. This will take you to the ORCID site and allow you to create a new iD or authenticate a pre-existing iD in Editorial Manager. 3. Thank you for stating the following financial disclosure:  [This study was supported by General Project of Shaanxi Provincial Department of Science and Technology- Social Development Field (grant number 2021SF-256). ].  Please state what role the funders took in the study.  If the funders had no role, please state: ""The funders had no role in study design, data collection and analysis, decision to publish, or preparation of the manuscript."" If this statement is not correct you must amend it as needed. Please include this amended Role of Funder statement in your cover letter; we will change the online submission form on your behalf. 4. Please include captions for your Supporting Information files at the end of your manuscript, and update any in-text citations to match accordingly. Please see our Supporting Information guidelines for more information: http://journals.plos.org/plosone/s/supporting-information. 

Reviewers' comments:

Reviewer's Responses to Questions

**Comments to the Author**

1. Is the manuscript technically sound, and do the data support the conclusions?

Reviewer #1: Yes

Reviewer #2: Partly

2. Has the statistical analysis been performed appropriately and rigorously? 

Reviewer #1: Yes

Reviewer #2: Yes

3. Have the authors made all data underlying the findings in their manuscript fully available?

Reviewer #1: Yes

Reviewer #2: Yes

4. Is the manuscript presented in an intelligible fashion and written in standard English?

Reviewer #1: Yes

Reviewer #2: Yes

5. Review Comments to the Author

Reviewer #1: The study showed that Lower serum bicarbonate levels are significantly associated with an increased 28-day mortality risk in ARDS patients, with particular emphasis on female patients, those with higher CCI scores, and those with milder ARDS. These findings indicate that monitoring and managing serum bicarbonate levels may be vital for improving survival in ARDS patients. I have a few comments:

1. Many studies have shown that sodium bicarbonate infusion can influence the serum level of bicarbonate (PMID: 30255318); The current work failed to address this, you need to mention and discuss this point.

2. The study lacks external validation and may not be applicable to Chinese population.

3. What is the causes of variation in bicarbonate in ARDS? is that caused by respiratory dysfunction or renal failure?

4. The heterogeneity of the study population should be acknowledged so that future work are needed to explore how subgroups of patients can have different results/conclusions (https://doi.org/10.1016/j.lers.2024.02.001). There has been numerous studies in this field and the authors may need to discuss this issue in interpreting current findings. 

5. "These findings indicate that monitoring and managing serum bicarbonate levels may be vital for improving survival in ARDS patients."---this conclusion cannot be inferred from current analysis because there is no causal exploration in the work.

6. The association of bicarbonate with mortality can be non-linear, you can model this in the multivariate equation.

Reviewer #2: Response Letter: Review of Manuscript PONE-D-24-58051

Title: Association between bicarbonate levels and mortality among acute respiratory distress syndrome patients: An analysis based on the Medical Information Mart for Intensive Care database

Dear authors,

Thank you for letting me review your work, and congratulations on it. I have attached the following comments:

The manuscript explores an important clinical question, investigating the association between serum bicarbonate levels and mortality in ARDS patients. The MIMIC-IV database is appropriate, and the methodology appears sound. However, several areas require clarification and improvement.

Specific Comments

1. Title and Abstract: The title is informative but could be clarified. The abstract provides a concise summary.

2. Introduction: While the introduction sets the context well, it could better highlight the theoretical background of the association of low bicarbonate and worse outcomes as already described in other critical illnesses.

3. Methods: The MIMIC-IV database needs to be cited. The methodology is detailed, and the statistical process is reasonable. Please elaborate on why you did not conduct a sensitivity analysis to assess the role of sodium bicarbonate administration in predicting mortality in ARDS. By excluding patients who received sodium bicarbonate or adjusting for its administration in multivariable models, the authors could assess the robustness of their findings regarding baseline bicarbonate levels. Without a sensitivity analysis, the results may overlook whether sodium bicarbonate use is a mediator, effect modifier, or confounder in the observed relationship between bicarbonate levels and mortality. Also, why did you not consider a propensity score matching?

4. Results: Several intriguing findings require further elaboration, particularly in the discussion section. The patient population comprises more males than females. What accounts for these gender differences? Are protective hormones a factor? What about the ARDS phenotypes?

What is the timing of bicarbonate level assessments in relation to the onset of ARDS? Bicarbonate levels do not have a definitive cutoff but exist within a range. Why did the authors choose to dichotomise this? Please provide additional details and consider referencing the normal range of 22-29 mmol/L, as it varies by source. Furthermore, the higher incidence of AKI in the non-survivors could influence bicarbonate levels and skew the results. Why do lower bicarbonate levels correlate with increased mortality in milder ARDS cases?

5. Discussion and Conclusion: The discussion provides a comprehensive overview, summarizing various studies that present differing conclusions on bicarbonate levels and their impact on mortality. However, starting from line 306, the discussion becomes somewhat repetitive.

I did not have access to the supplementary materials.

Overall, the presented study addresses a clinically significant question with potential implications for ARDS management. The large sample size enhances the robustness of the findings.

However, some aspects of the methodology lack clarity.

6. PLOS authors have the option to publish the peer review history of their article (what does this mean? ). If published, this will include your full peer review and any attached files.

**Do you want your identity to be public for this peer review?** For information about this choice, including consent withdrawal, please see our Privacy Policy .

Reviewer #1: No

Reviewer #2: No

---

## [Author Response · Author response to Decision Letter 1]

30 Mar 2025

Comments to the Author

1. Is the manuscript technically sound, and do the data support the conclusions?

Reviewer #1: Yes

Reviewer #2: Partly

Reply: Thanks for your comment.

2. Has the statistical analysis been performed appropriately and rigorously?

Reviewer #1: Yes

Reviewer #2: Yes

Reply: Thanks for your comment.

3. Have the authors made all data underlying the findings in their manuscript fully available?

Reviewer #1: Yes

Reviewer #2: Yes

Reply: Thanks for your comment.

4. Is the manuscript presented in an intelligible fashion and written in standard English?

Reviewer #1: Yes

Reviewer #2: Yes

Reply: Thanks for your comment.

5. Review Comments to the Author

Reviewer #1: The study showed that Lower serum bicarbonate levels are significantly associated with an increased 28-day mortality risk in ARDS patients, with particular emphasis on female patients, those with higher CCI scores, and those with milder ARDS. These findings indicate that monitoring and managing serum bicarbonate levels may be vital for improving survival in ARDS patients. I have a few comments:

1. Many studies have shown that sodium bicarbonate infusion can influence the serum level of bicarbonate (PMID: 30255318); The current work failed to address this, you need to mention and discuss this point.

Reply: Thanks for your comment. In the multivariate Cox proportional hazards model, we adjusted a series of confounding factors to explore the association between baseline bicarbonate levels with the 28-day mortality in ARDS patients, including bicarbonate infusion. Furthermore, we conducted subgroup analyses specifically examining the association between baseline bicarbonate levels and 28-day mortality in ARDS patients based on their bicarbonate infusion status. Notably, we observed in subgroup without bicarbonate infusion, lower baseline bicarbonate levels were still associated with high 28-day mortality in ARDS patients.

2. The study lacks external validation and may not be applicable to Chinese population.

Reply: Thanks for your comment. In this study, the data on patients with ARDS admitted to the ICU were extracted from the MIMIC-IV database to assess the association between bicarbonate levels and the risk of 28-day mortality. Our study was derived from a single-center database, which may have limitations such as limited external validity, bias in selection and results, and insufficient data diversity. While acknowledging these limitations in the Discussion section of the manuscript, we also look forward to future research directions to enhance the generalizability of our findings:

However, we have to acknowledge that our study has some objective limitations. First, this was a retrospective study with observational data, precluding the definitive establishment of causality relationship between the 28-day mortality in patients with ARDS and their baseline bicarbonate levels. Nonetheless, we established univariate and multivariate Cox proportional hazards models and subgroup analyses based on specific characteristics of the study population to yield and validate robust and credible results. Second, the data of present study were only derived from the MIMI-IV database. Although this database has large samples and contains rich clinical information, single-center study inevitably has shortcomings such as limited external validity, heterogeneity of the study population, specificity of medical institutions, and insufficient diversity of data. Therefore, caution is needed when generalizing our observations to patient populations across different regions, ethnicities, socioeconomic backgrounds, or medical level. Certainly, it is essential to optimize inclusion and exclusion criteria to avoid excessive heterogeneity in the enrolled population and to perform additional subgroup analyses to mitigate the impact of population diversity on study outcomes. Third, although we controlled for as many potential confounders as possible, residual bias may persist due to limitations in data availability within MIMIC -IV and unaccounted confounding factors-particularly our inability to discern whether bicarbonate was administered intravenously or orally. Forth, the bicarbonate levels only applied to patients admitted to the ICU, which limited our ability to assess how fluctuations in bicarbonate after ICU stay may impact the mortality, potentially affecting the accuracy of our results. Therefore, while the study provides important preliminary evidence, well-designed, multicenter prospective studies or randomized controlled trials (RCTs) are needed to validate these findings and assess their broader applicability.

3. What are the causes of variation in bicarbonate in ARDS? is that caused by respiratory dysfunction or renal failure?

Reply: Thanks for your comment. In ARDS patients, changes in bicarbonate (HCO₃⁻) levels are typically driven by multiple pathophysiological mechanisms, primarily involving acid-base disturbances, compensatory responses, and organ dysfunction. Among these, renal failure is one of the central factors contributing to HCO₃⁻ dysregulation in ARDS patients, primarily disrupting acid-base balance through the following mechanisms:

1.Loss of proximal tubule acidification capacity: In AKI, injury to proximal tubular epithelial cells leads to decreased activity of the Na⁺-H⁺ exchanger (NHE3) and carbonic anhydrase (CA), impairing HCO₃⁻ reabsorption.

2.Accumulation of acidic metabolites: During renal failure, reduced excretion of non-volatile acids (e.g., sulfates, phosphates, uric acid) depletes HCO₃⁻ buffers, manifesting as an elevated anion gap (AG) with normal or mildly elevated lactate levels.

3.Synergistic injury between ARDS and AKI: High positive end expiratory pressure (PEEP) ventilation in ARDS patients reduces venous return, causing renal hypoperfusion that ultimately exacerbates AKI.

4.Amplification by inflammatory mediators: Pro-inflammatory cytokines (e.g., IL-6, TNF-α) released during ARDS directly damage renal tubular epithelial cells.

4. The heterogeneity of the study population should be acknowledged so that future work is needed to explore how subgroups of patients can have different results/conclusions (https://doi.org/10.1016/j.lers.2024.02.001). There have been numerous studies in this field and the authors may need to discuss this issue in interpreting current findings. 

Reply: Thanks for your comments. We have acknowledged the limitations of this study in the Discussion section of our research, including the heterogeneity of the study population. Population heterogeneity represents significant differences in baseline characteristics, disease presentation, treatment response, or risk of outcome among individuals studied. In the current study, we analyzed analyses in groups by age, gender, CCI, ARDS severity, and with or without bicarbonate administration. However, further large, well-designed, multicenter prospective studies are needed to avoid over-broadening the inclusion population by optimizing the inclusion and exclusion criteria, and to conduct more subgroup analyses to address the impact of heterogeneity in the study population on the results.

5. "These findings indicate that monitoring and managing serum bicarbonate levels may be vital for improving survival in ARDS patients."-this conclusion cannot be inferred from current analysis because there is no causal exploration in the work.

Reply: Thanks for your comments. We realized that based on our current study, this conclusion may be an overreach. We have faithfully described the clinical reference value of this study based on the results observed in this study:

Baseline bicarbonate levels of ARDS patients in ICU have certain clinical reference value for the development of clinical management and the assessment of prognostic risk during the ICU admission.

6. The association of bicarbonate with mortality can be non-linear, you can model this in the multivariate equation.

Reply: Thanks for your comment. The restricted cubic splines (RCS) analysis was conducted to explore the non-linear association between the bicarbonate levels and mortality. The result of RCS was exhibited in below:

The RCS curve analysis revealed a nonlinear relationship between the bicarbonate levels and 28-day mortality in patients with ARDS (P for non-linear <0.001).

All changed were demonstrated in the manuscript.

Reviewer #2: Response Letter: Review of Manuscript PONE-D-24-58051

Title: Association between bicarbonate levels and mortality among acute respiratory distress syndrome patients: An analysis based on the Medical Information Mart for Intensive Care database

Dear authors,

Thank you for letting me review your work, and congratulations on it. I have attached the following comments:

The manuscript explores an important clinical question, investigating the association between serum bicarbonate levels and mortality in ARDS patients. The MIMIC-IV database is appropriate, and the methodology appears sound. However, several areas require clarification and improvement.

Reply: Thank you for your recognition of our work. We have revised the manuscript by point to point according to your valuable suggestions, and hope that the revised manuscript will meet the requirements of the PLoS One publication.

Specific Comments

1. Title and Abstract: The title is informative but could be clarified. The abstract provides a concise summary.

Reply: Thanks for your comments. We have polished the Abstract. We hope that we can effectively convey the value of our research with clear structure and precise language.

2. Introduction: While the introduction sets the context well, it could better highlight the theoretical background of the association of low bicarbonate and worse outcomes as already described in other critical illnesses.

Reply: Thanks for your comments. We have described in detail the physiological function of bicarbonate and the potential physiological mechanism between bicarbonate and endothelial dysfunction and ARDS. Meanwhile, we further described the association of bicarbonate with the prognosis of other critically ill patients.

3. Methods: The MIMIC-IV database needs to be cited. The methodology is detailed, and the statistical process is reasonable. Please elaborate on why you did not conduct a sensitivity analysis to assess the role of sodium bicarbonate administration in predicting mortality in ARDS. By excluding patients who received sodium bicarbonate or adjusting for its administration in multivariable models, the authors could assess the robustness of their findings regarding baseline bicarbonate levels. Without a sensitivity analysis, the results may overlook whether sodium bicarbonate use is a mediator, effect modifier, or confounder in the observed relationship between bicarbonate levels and mortality. Also, why did you not consider a propensity score matching?

Reply: Thanks for your comments.

First, we have cited the official link in the manuscript to the source of the data for the MIMIC-IV database.

Second, the present study population included 5,917 ARDS patients with non-bicarbonate administration and 460 ARDS patients with bicarbonate administration. Sensitivity analyses were conducted to further explore the association between baseline bicarbonate levels and their 28-day mortality risk in patients with and without bicarbonate administration. The results were shown in below Table:

Variates HR (95%CI) P

Non-bicarbonate administration (n=5917)

Baseline bicarbonate level 0.98 (0.96-0.99) 0.006

Baseline bicarbonate level

<23.00 1.26 (1.07-1.48) 0.004

≥23.00 Ref

Bicarbonate administration (n=460)

Baseline bicarbonate level 1.01 (0.97-1.05) 0.687

Baseline bicarbonate level

<23.00 0.88 (0.56-1.38) 0.575

≥23.00 Ref

Notes: HR: hazard ratio; CI: confidence intervals; Ref: reference;

Adjusted age, gender, race; marital status; weight; diastolic, respiratory rate, SOFA, SAPSII, CCI, platelet; lactate; magnesium; sodium; chloride, RRT, propofol, dexmedetomidine, opioid; and antibiotics.

Results of sensitivity analyses showed that lower baseline bicarbonate levels in ARDS patients without bicarbonate administration during their ICU stay were significantly associated with a higher 28-day risk of mortality.

First, PSM excludes unmatched individuals, resulting in reduced sample size, which may introduce selection bias and diminish statistical power. Second, the current study specifically evaluated both linear and non-linear effects of the exposure variable (serum bicarbonate) as a continuous measure, rather than a binary dichotomy. Consequently, PSM was deemed methodologically inappropriate for this analytical context.

4.Results: Several intriguing findings require further elaboration, particularly in the discussion section. The patient population comprises more males than females. What accounts for these gender differences? Are protective hormones a factor? What about the ARDS phenotypes?

Reply: Thanks for your comments. We included a total of 6,377 patients with ARDS, including 2,271 women and 4,106 men, extracted from the MIMIC-IV database after a series of inclusion and exclusion treatments. The MIMIC-IV database is a large critical care medicine database maintained by the Laboratory of Computational Physiology at the Massachusetts Institute of Technology (MIT) that contains real-world clinical data from Beth Israel Deaconess Medical Center in Boston, USA. Our data show the true gender distribution of ARDS patients admitted to the ICU between 2008 and 2019. The primary objective of our study was to investigate the association between baseline bicarbonate levels and the 28-day mortality risk in patients with ARDS during ICU admission, with gender being one of the important demographic confounders. However, our study is observational, and further large-scale studies related to the underlying physiological mechanisms are needed in the future. Gender differences are an interesting area of future research, and we look forward to more research to uncover the role of gender in association between bicarbonate levels and the risk of short-term prognosis for ARDS.

What is the timing of bicarbonate level assessments in relation to the onset of ARDS? Bicarbonate levels do not have a definitive cutoff but exist within a range. Why did the authors choose to dichotomise this? Please provide additional details and consider referencing the normal range of 22-29 mmol/L, as it varies by source. Furthermore, the higher incidence of AKI in the non-survivors could influence bicarbonate levels and skew the results. Why do lower bicarbonate levels correlate with increased mortality in milder ARDS cases?

Reply: Thanks for your comment. First, we looked at the initial bicarbonate level gathered an I

---

## [Decision Letter · Decision Letter 1]

29 Apr 2025

PONE-D-24-58051R1Association between bicarbonate levels and mortality among acute respiratory distress syndrome patients: An analysis based on Medical Information Mart for Intensive Care databasePLOS ONE

Dear Dr. Han,

Thank you for submitting your manuscript to PLOS ONE. After careful consideration, we feel that it has merit but does not fully meet PLOS ONE’s publication criteria as it currently stands. Therefore, we invite you to submit a revised version of the manuscript that addresses the points raised during the review process.

**Introduction: **

While the term endothelial dysfunction is introduced in line 55, a brief explanation of what it entails, specifically in the context of ARDSIn lines 56-61, you mention that low bicarbonate levels reflect metabolic acidosis and that this is a contributing factor to poor outcomes in ARDS patients. Consider specifying how bicarbonate directly impacts acid-base balance and how metabolic acidosis worsens ARDS.emphasize the novelty of the study a bit more.The objective of the study is well-stated, but could be more precise in how it will help clinical decision-making

**Methods:**

Add s brief note on the validation of the MIMIC-IV database or any potential limitations specific to the dataset that might affect the study's conclusions.Expand to specify how the data was structured (e.g., patient demographics, medical histories, treatment details)

The statement "ethical approval from the Second Affiliated Hospital of Xi'an Jiaotong University was waived" needs additional clarificationThe term "meticulously" in "meticulously extracted" could be omitted or replaced with more objective language, as "meticulously" is somewhat subjective. Consider using "systematically" or "using predefined codes."Briefly mention the specific criteria used in the Berlin definition of ARDSClarify the purpose of each statistical model and analysis, especially the difference between univariate and multivariable Cox models.

Expand on how sensitivity analysis was performed and why it was necessary, especially the comparison between datasets before and after imputation.The statement "detailed results are presented in Supplementary Table 1" need more context about the importance of the table or a brief description of what it contains to guide readers.

The description of the multiple imputation process is generally clear; however, the reference or package name is missing after "mice()" — please insert the appropriate package name or a proper citation inside the parentheses.The choice of the 65-year cutoff for subgrouping by age should be justified, either by citing clinical relevance or previous literature.

**Discussion**

The discussion presents a large number of studies with repetition of the main finding several times (e.g., lines 268–272, 326–328, 342–344, 384–386) back-to-back without a strong thematic integration. This can overwhelm the reader and obscure your main findings. Group the studies thematically: For example, start by discussing studies on bicarbonate levels in non-ARDS populations (e.g., cancer, dialysis, stroke), then transition into critically ill populations, and finally position your study within this context.Use linking sentences between studies to explain how each cited work builds toward the necessity of your study.

There’s a very long list of studies about bicarbonate levels in different diseases (lines 273–313), but only some of them are truly relevant to ARDS. The link to ARDS literature is weak. Most cited studies are on *CKD* , *dialysis* , *cancer* , or *stroke* patients, not ARDS patients.

Bicarbonate treatment is mentioned in subgroups but is not deeply discussed. Briefly discuss whether bicarbonate supplementation has clear evidence in ARDS treatment or remains controversial.You mention that this is the first study to address bicarbonate and 28-day mortality in ARDS patients using the MIMIC-IV database. Restate the gap clearly at the beginning and end of the discussion and explain why this gap matters: Why would clinicians or researchers care about bicarbonate monitoring in ARDS?

Ensure that all important claims are properly cited. Why females? Why mild ARDS? Some speculation is offered (gender physiology, compensation), Add citations for statements.

The Limitations section is thorough but could be more concise. Merge similar points (single-center study, limited generalizability, need for external validation) into a tighter paragraph. Acknowledge potential reverse causality: Lower bicarbonate could be a marker of severity rather than a direct cause. Discuss missing variables: For instance, lactate levels or detailed ventilator parameters could confound the bicarbonate-mortality relationship.

Minor typos: "forth" (should be "fourth") at line 377.Ensure all proper nouns and terms (e.g., ARDS, ICU) are consistently capitalized.Phrasing: "definitive establishment of causality relationship" to "definitive establishment of a causal relationship."Some sentences are quite lengthy and could benefit from splitting for clarity. E.g., Line 358: "The patients’ mortality risk could be driven by several mechanisms…"

We look forward to receiving your revised manuscript.

Kind regards,

Reema Karasneh

Academic Editor

PLOS ONE

Journal Requirements:

Additional Editor Comments:

Introduction:

• While the term endothelial dysfunction is introduced in line 55, a brief explanation of what it entails, specifically in the context of ARDS

• In lines 56-61, you mention that low bicarbonate levels reflect metabolic acidosis and that this is a contributing factor to poor outcomes in ARDS patients. Consider specifying how bicarbonate directly impacts acid-base balance and how metabolic acidosis worsens ARDS.

• emphasize the novelty of the study a bit more.

• The objective of the study is well-stated, but could be more precise in how it will help clinical decision-making

Methods:

• Add s brief note on the validation of the MIMIC-IV database or any potential limitations specific to the dataset that might affect the study's conclusions.

• Expand to specify how the data was structured (e.g., patient demographics, medical histories, treatment details)

• The statement "ethical approval from the Second Affiliated Hospital of Xi'an Jiaotong University was waived" needs additional clarification

• The term "meticulously" in "meticulously extracted" could be omitted or replaced with more objective language, as "meticulously" is somewhat subjective. Consider using "systematically" or "using predefined codes."

• Briefly mention the specific criteria used in the Berlin definition of ARDS

• Clarify the purpose of each statistical model and analysis, especially the difference between univariate and multivariable Cox models.

• Expand on how sensitivity analysis was performed and why it was necessary, especially the comparison between datasets before and after imputation.

• The statement "detailed results are presented in Supplementary Table 1" need more context about the importance of the table or a brief description of what it contains to guide readers.

• The description of the multiple imputation process is generally clear; however, the reference or package name is missing after "mice()" — please insert the appropriate package name or a proper citation inside the parentheses.

• The choice of the 65-year cutoff for subgrouping by age should be justified, either by citing clinical relevance or previous literature.

Discussion

• The discussion presents a large number of studies with repetition of the main finding several times (e.g., lines 268–272, 326–328, 342–344, 384–386) back-to-back without a strong thematic integration. This can overwhelm the reader and obscure your main findings. Group the studies thematically: For example, start by discussing studies on bicarbonate levels in non-ARDS populations (e.g., cancer, dialysis, stroke), then transition into critically ill populations, and finally position your study within this context.

• Use linking sentences between studies to explain how each cited work builds toward the necessity of your study.

• There’s a very long list of studies about bicarbonate levels in different diseases (lines 273–313), but only some of them are truly relevant to ARDS. The link to ARDS literature is weak. Most cited studies are on CKD, dialysis, cancer, or stroke patients, not ARDS patients.

• Bicarbonate treatment is mentioned in subgroups but is not deeply discussed. Briefly discuss whether bicarbonate supplementation has clear evidence in ARDS treatment or remains controversial.

• You mention that this is the first study to address bicarbonate and 28-day mortality in ARDS patients using the MIMIC-IV database. Restate the gap clearly at the beginning and end of the discussion and explain why this gap matters: Why would clinicians or researchers care about bicarbonate monitoring in ARDS?

• Ensure that all important claims are properly cited. Why females? Why mild ARDS? Some speculation is offered (gender physiology, compensation), Add citations for statements.

• The Limitations section is thorough but could be more concise. Merge similar points (single-center study, limited generalizability, need for external validation) into a tighter paragraph. Acknowledge potential reverse causality: Lower bicarbonate could be a marker of severity rather than a direct cause. Discuss missing variables: For instance, lactate levels or detailed ventilator parameters could confound the bicarbonate-mortality relationship.

• Minor typos: "forth" (should be "fourth") at line 377.

• Ensure all proper nouns and terms (e.g., ARDS, ICU) are consistently capitalized.

• Phrasing: "definitive establishment of causality relationship" to "definitive establishment of a causal relationship."

• Some sentences are quite lengthy and could benefit from splitting for clarity. E.g., Line 358: "The patients’ mortality risk could be driven by several mechanisms…"

Reviewers' comments:

Reviewer's Responses to Questions

**Comments to the Author**

1. If the authors have adequately addressed your comments raised in a previous round of review and you feel that this manuscript is now acceptable for publication, you may indicate that here to bypass the “Comments to the Author” section, enter your conflict of interest statement in the “Confidential to Editor” section, and submit your "Accept" recommendation.

Reviewer #1: (No Response)

Reviewer #2: All comments have been addressed

2. Is the manuscript technically sound, and do the data support the conclusions?

Reviewer #1: (No Response)

Reviewer #2: Yes

3. Has the statistical analysis been performed appropriately and rigorously? 

Reviewer #1: (No Response)

Reviewer #2: Yes

4. Have the authors made all data underlying the findings in their manuscript fully available?

Reviewer #1: (No Response)

Reviewer #2: Yes

5. Is the manuscript presented in an intelligible fashion and written in standard English?

Reviewer #1: (No Response)

Reviewer #2: Yes

6. Review Comments to the Author

Reviewer #1: my previous comments are well addressed. my previous comments are well addressed. my previous comments are well addressed. my previous comments are well addressed.

Reviewer #2: Thank you for addressing my comments and concerns. I think the manuscript at its current form is publishable.

7. PLOS authors have the option to publish the peer review history of their article (what does this mean? ). If published, this will include your full peer review and any attached files.

**Do you want your identity to be public for this peer review?** For information about this choice, including consent withdrawal, please see our Privacy Policy .

Reviewer #1: No

Reviewer #2: No

---

## [Author Response · Author response to Decision Letter 2]

2 May 2025

Introduction:

• While the term endothelial dysfunction is introduced in line 55, a brief explanation of what it entails, specifically in the context of ARDS.

Reply: Thanks for your comment. In describing the pathological basis of patients with ARDS, we further describe the association of endothelial dysfunction with the occurrence and progression of ARDS: Previous clinical studies have revealed that ARDS is characterized by inflammatory pulmonary edema and impaired gas exchange, with endothelial injury serving as one of the pivotal mechanisms underlying its pathogenesis and progression. Endothelial injury drives the pathological process of ARDS by increasing microvascular permeability, disrupting the coagulation-fibrinolysis balance, and amplified inflammatory responses.

• In lines 56-61, you mention that low bicarbonate levels reflect metabolic acidosis and that this is a contributing factor to poor outcomes in ARDS patients. Consider specifying how bicarbonate directly impacts acid-base balance and how metabolic acidosis worsens ARDS.

Reply: Thanks for your comment. We systematically elucidated the association between bicarbonate levels and systemic acid-base balance, along with the specific pathways through which metabolic acidosis exacerbates ARDS The bicarbonate buffer system serves as one of the most primary buffer systems for maintaining acid-base balance in the human body. When acidic substances increase, bicarbonate can combine with hydrogen ions to form carbon dioxide and water, thereby neutralizing the acids. Excessive loss bicarbonate compromises the body’s ability to neutralize acid, leading to accumulation of acidic substances in extracellular fluid and subsequent metabolic acidosis. Metabolic acidosis aggravates ARDS by disrupting the alveolar-capillary barrier, increasing endothelial injury and amplifying inflammatory responses

• emphasize the novelty of the study a bit more.

Reply: Thanks for your comment. We have elaborated on the novelty and clinical significance of this article in the Introduction and Discussion sections: Understanding this relationship may provide valuable insights into risk stratification and the potential role of bicarbonate levels as a prognostic marker, contributing to more clinical decision-making and tailored management strategies for ARDS patients.

• The objective of the study is well-stated, but could be more precise in how it will help clinical decision-making

Reply: Thanks for your comment. In the introduction section, we briefly described the potential clinical significance of this study. In the discussion part, we elaborated in detail on the clinical significance of using baseline bicarbonate level as a potential prognostic indicator for patients with ARDS.

Baseline bicarbonate level, as a potential prognostic biomarker for ARDS, holds dual clinical significance: on the one hand, lower bicarbonate levels reflect the metabolic acidosis and is associated with multi-organ dysfunction; on the other hand, lower bicarbonate levels can provide a reference for the stratification of mortality risk and help guide more effective clinical management of patients with ARDS in the ICU.

Methods:

• Add s brief note on the validation of the MIMIC-IV database or any potential limitations specific to the dataset that might affect the study's conclusions.

Reply: Thanks for your comment. We have added the verification information for the MIMIC- IV database: To access this database, one of the authors (Junli Han) obtained the necessary certification and subsequently extracted the relevant variables for our study (certification number: 13804349). We have elaborated on the limitations of the MIMIC-IV database in the discussion section.

• Expand to specify how the data was structured (e.g., patient demographics, medical histories, treatment details)

Reply: Thanks for your comment. In the data extraction section of the manuscript, we have presented the clinical data extracted in this study in detail by category such as demographic information, comorbidities, vital signs, and treatment modalities. Furthermore, we have also re-presented the baseline table in accordance with such categories.

• The statement "ethical approval from the Second Affiliated Hospital of Xi'an Jiaotong University was waived" needs additional clarification.

Reply: Thanks for your comment. We have re-stated the ethical approval for this study: The use of the MIMIC-IV database was approved by the review committee of Massachusetts Institute of Technology and Beth Israel Deaconess Medical Center� therefore, the ethics review committee of the Second Affiliated Hospital of Xi’an Jiaotong University waived the ethics approval statement and the requirement for informed consent for this study.

• The term "meticulously" in "meticulously extracted" could be omitted or replaced with more objective language, as "meticulously" is somewhat subjective. Consider using "systematically" or "using predefined codes."

Reply: Thanks for your comment. We have revised “meticulously” to “systematically”.

• Briefly mention the specific criteria used in the Berlin definition of ARDS.

Reply: Thanks for your comment. We have already described the Berlin definition of ARDS and further described the definitions of mild, moderate and severe ARDS: ARDS was defined according to the Berlin criteria, which include the following: acute onset, a PaO2/FiO2 ≤300 mmHg, PEEP ≥5 cm H2O, bilateral infiltrates on chest radiograph, and the absence of heart failure. ARDS severity was classified based on the PaO2/FiO2 ratio: mild (200 mmHg <PaO2/FiO2 ≤300 mmHg), moderate (100 mmHg <PaO2/FiO2 ≤200 mmHg), and severe (PaO2/FiO2 ≤100 mmHg).

• Clarify the purpose of each statistical model and analysis, especially the difference between univariate and multivariable Cox models.

Reply: Thanks for your comment. We have described in detail the purpose of implementing each model and analysis in the statistical analysis section. Among them, the univariate Cox hazard proportional model was used to preliminarily screen the covariates with statistical association with the 28-day mortality risk of ARDS patients, excluding non-statistically significant variables, and reducing the dimension of subsequent multivariate analysis. The covariates with statistically significant associations with outcomes selected from the univariate Cox proportional hazards model analysis were included in the multivariate Cox proportional hazards model analysis, and the association between bicarbonate levels and the 28-day mortality risk of ARDS patients was clarified by controlling for confounding.

• Expand on how sensitivity analysis was performed and why it was necessary, especially the comparison between datasets before and after imputation.

Reply: Thanks for your comment. Sensitivity analysis of missing data before and after imputation is a critical step in clinical research to ensure the reliability of results. We excluded variables with missing value >20%, and variables with missing value ≤20% were multi-imputed. In our study, we describe in detail the method of multiple imputation, and the purpose of the sensitivity analysis before and after multiple imputation: that is, considering the potential impact of this multiple imputation method on the relationship between exposure and outcome, we performed a sensitivity analysis comparing the results before and after multiple imputation to confirm the robustness of the model and exclude the effect of random forest imputation on the results.

• The statement "detailed results are presented in Supplementary Table 1" need more context about the importance of the table or a brief description of what it contains to guide readers.

Reply: Thanks for your comment. Supplementary Table 1 presented the sensitivity analysis before and after the imputation of missing data. We elaborated on the results of Supplementary Table 1 in the manuscript: The distribution differences of all variables before and after imputation were not statistically significant (all P>0.05), thus, the imputation of missing data conducted in this study supported the robustness of the study results.

• The description of the multiple imputation process is generally clear; however, the reference or package name is missing after "mice()" — please insert the appropriate package name or a proper citation inside the parentheses.

Reply: Thanks for your comment. The parentheses indicate this is the standard function call syntax in R programming, where:

• mice refer to the Multiple Imputation by Chained Equations algorithm;

• Empty parentheses () denote the function's default parameter structure

We have made the following revisions in the manuscript Missing values were handled using the mice () function from R’s mice package (v 4.3.1), which starts with a data frame containing missing values and generates multiple (default of five) complete datasets by imputing the missing values.

• The choice of the 65-year cutoff for subgrouping by age should be justified, either by citing clinical relevance or previous literature.

Reply: Thanks for your comment. To assess the robustness of the association between baseline bicarbonate levels and 28-day mortality risk in ARDS patients, we stratified the study population into two subgroups: those aged ≥65 years and those <65 years. The rationale for selecting 65 years as the cutoff was based on the following considerations: first, from the perspective of biological Plausibility, 65 years is a well-established critical threshold for "immunosenescence," characterized by significant decline in alveolar epithelial repair capacity and more severe dysregulation of inflammatory responses; second, The Berlin Definition classifies older ARDS patients as those aged ≥65 years, reflecting distinct pathophysiological and prognostic features in this population; finally, in the U.S., 65 years marks the eligibility age for Medicare coverage, which may influence treatment intensity decisions in clinical practice.

In summary, we conducted stratified analyses by age (≥65 vs. <65 years) to further investigate the relationship between baseline bicarbonate levels and 28-day mortality risk across different age groups in ARDS patients.

Discussion

• The discussion presents a large number of studies with repetition of the main finding several times (e.g., lines 268–272, 326–328, 342–344, 384–386) back-to-back without a strong thematic integration. This can overwhelm the reader and obscure your main findings. Group the studies thematically: For example, start by discussing studies on bicarbonate levels in non-ARDS populations (e.g., cancer, dialysis, stroke), then transition into critically ill populations, and finally position your study within this context. Use linking sentences between studies to explain how each cited work builds toward the necessity of your study.

Reply: Thanks for your comment. Thank you very much for your valuable suggestions. We have reorganized the discussion part in accordance with your opinions. All the revisions are presented in red marks in the discussion section of the manuscript.

• There’s a very long list of studies about bicarbonate levels in different diseases (lines 273–313), but only some of them are truly relevant to ARDS. The link to ARDS literature is weak. Most cited studies are on CKD, dialysis, cancer, or stroke patients, not ARDS patients.

Reply: Thanks for your comment. Current clinical research on the association between bicarbonate levels and short-term prognosis in ARDS patients remains limited. In our discussion section, we initially reviewed existing evidence regarding bicarbonate levels and outcomes in various diseases. This approach served two purposes:

1. It highlighted the significance and novelty of our study by contextualizing it within the substantial disease burden of ARDS;

2. It provided clinical research references for investigating the bicarbonate-ARDS prognosis relationship.

Subsequently, we focused on explaining the potential physiological mechanisms underlying the association between bicarbonate levels and short-term outcomes in ARDS patients. In response to your valuable suggestion, we have substantially reduced the discussion of bicarbonate's role in other diseases, shifting our emphasis to interpreting the specific findings of our current study.

• Bicarbonate treatment is mentioned in subgroups but is not deeply discussed. Briefly discuss whether bicarbonate supplementation has clear evidence in ARDS treatment or remains controversial.

Reply: Thanks for your comment. We conducted subgroup analyses of patients with ARDS with different characteristics, and in the discussion, we attempted to explain the results of the subgroups in terms of physiological mechanisms. However, the focus of our study was to explore the association between baseline bicarbonate levels and the 28-day mortality risk in patients with ARDS, and the aim of subgroup analyses was to demonstrate that this association remained robust in patients with ARDS with different characteristics, and to increase some confidence in the association between exposure and outcomes. We also stated in the article that the association between baseline bicarbonate levels and short-term outcomes in patients with ARDS with different characteristics still needs further specialized studies to unravel. In addition, we looked at the association between baseline bicarbonate levels and their short-term mortality risk in patients with ARDS admitted to the ICU, rather than the association between bicarbonate use and their prognosis. Of course, the association between bicarbonate supplementation and the prognosis of ARDS patients is a very valuable research direction, and I hope that we or other researchers can further pay attention to the association between bicarbonate use and the prognosis of ARDS patients in the future, to provide more insights for clinicians to carry out accurate medical management of ARDS patients.

• You mention that this is the first study to address bicarbonate and 28-day mortality in ARDS patients using the MIMIC-IV database. Restate the gap clearly at the beginning and end of the discussion and explain why this gap matters: Why would clinicians or researchers care about bicarbonate monitoring in ARDS?

Reply: Thanks for your comment. We further elaborated on the clinical significance of our study in the discussion section.

One of its major strengths was that it is the first to investigate the association between baseline bicarbonate levels and short-term mortality risk in critically ill ARDS patients, using a large-scale cohort. Baseline bicarbonate level, as a potential prognostic biomarker for ARDS, holds dual clinical significance: on the one hand, lower bicarbonate levels reflect the metabolic acidosis and is associated with multi-organ dysfunction; on the other hand, lower bicarbonate levels can provide a reference for the stratification of mortality risk and help guide more effective clinical management of patients with ARDS in the ICU.

• Ensure that all important claims are properly cited. Why females? Why mild ARDS? Some speculation is offered (gender physiology, compensation), Add citations for statements.

Reply: Thanks for your comment.

First, as clarified in our initial response to the reviewers, the ARDS patients included in this study reflect the real-world gender distribution of the MIMIC-IV database.

Second, in subsequent subgroup analyses stratified by patient characteristics, we observed that the association between baseline bicarbonate levels and 28-day mortality risk was more pronounced in female ARDS patients compared to males. To interpret this gender-based disparity, we propose in the Discussion that: Females may exhibit distinct physiological responses to metabolic acidosis or acid-base regulatory mechanisms, potentially rendering them more vulnerable to the adverse effects of low bicarbonate levels, thereby increasing mortality risk.

Similarly, subgroup results indicated that the bicarbonate-mortality association was stronger in patients with higher CCI scores and milder ARDS severity. A higher CCI is indicative of multiple comorbidities, which further complicates the management of ARDS and places an additional burden on

---

## [Editor Report · Decision Letter 2]

12 May 2025

PONE-D-24-58051R2Association between bicarbonate levels and mortality among acute respiratory distress syndrome patients: An analysis based on Medical Information Mart for Intensive Care databasePLOS ONE

Dear Dr. Han,

Thank you for submitting your manuscript to PLOS ONE. After careful consideration, we feel that it has merit but does not fully meet PLOS ONE’s publication criteria as it currently stands. Therefore, we invite you to submit a revised version of the manuscript that addresses the points raised during the review process.

Methods**: **

Typo at line 137: "eGFR was categorized a 137 s <60..." → Should be: "eGFR was categorized as <60, 60–90, or ≥90 mL/min/1.73 m²."Line 114-115: Formatting issue with “, therefore,” and "Second Affiliated Hospital..."

Discussion: paragraph lines [349-361] needs supporting references

We look forward to receiving your revised manuscript.

Kind regards,

Reema Karasneh

Academic Editor

PLOS ONE

Journal Requirements:

Additional Editor Comments (if provided):

**Methods: **

Typo at line 137: "eGFR was categorized a 137 s <60..." → Should be: "eGFR was categorized as <60, 60–90, or ≥90 mL/min/1.73 m²."Line 114-115: Formatting issue with “, therefore,” and "Second Affiliated Hospital..."

Discussion: paragraph lines [349-361] needs supporting references

---

## [Author Response · Author response to Decision Letter 3]

12 May 2025

Methods:

-Typo at line 137: "eGFR was categorized a 137 s <60..." → Should be: "eGFR was categorized as <60, 60–90, or ≥90 mL/min/1.73 m²."

Reply: Thanks for your comment. We have revised this typo.

-Line 114-115: Formatting issue with “, therefore,” and "Second Affiliated Hospital..."

Reply: Thanks for your comment. We have revised this typo.

Discussion: paragraph lines [349-361] needs supporting references.

Reply: Thans for your comment. We have added the supporting references for this section.

---

## [Editor Report · Decision Letter 3]

15 May 2025

Association between bicarbonate levels and mortality among acute respiratory distress syndrome patients: An analysis based on Medical Information Mart for Intensive Care database

PONE-D-24-58051R3

Dear Dr. Han,

We’re pleased to inform you that your manuscript has been judged scientifically suitable for publication and will be formally accepted for publication once it meets all outstanding technical requirements.

Kind regards,

Reema Karasneh

Academic Editor

PLOS ONE
---

## [Editor Report · Acceptance letter]

PONE-D-24-58051R3

PLOS ONE

Dear Dr. Han,

I'm pleased to inform you that your manuscript has been deemed suitable for publication in PLOS ONE. Congratulations! Your manuscript is now being handed over to our production team.

Kind regards,

on behalf of

Dr. Reema Karasneh

Academic Editor

PLOS ONE